# Changes in the Available Potential and Kinetic Energy of Mesoscale Vortices When They Are Stretched into Filaments

**Vladimir V. Zhmur** [1,2,3]**, Tatyana V. Belonenko** [3,]*****, Vladimir S. Travkin** [3,4]**, Elena V. Novoselova** [3]**,
David A. Harutyunyan** [2] **and Roshin P. Raj** [5,6]

1   Shirshov Institute of Oceanology, Russian Academy of Sciences, 117997 Moscow, Russia;
    zhmur-vladimir@mail.ru
2   Moscow Institute of Physics and Technology, 141701 Moscow, Russia; arutyunyan.da@phystech.edu
3   Institute of Earth Sciences, St. Petersburg State University, 199178 St. Petersburg, Russia;
    vtravkin99@gmail.com (V.S.T.); novoselovaa.elena@gmail.com (E.V.N.)
4   State Oceanographic Institute, 119034 Moscow, Russia
5   Nansen Environmental and Remote Sensing Center, N-5007 Bergen, Norway; roshin.raj@nersc.no
6   Bjerknes Center for Climate Research, University of Bergen, N-5020 Bergen, Norway
*   Correspondence: btvlisab@yandex.ru

**Abstract:** The article discusses various aspects of the interaction of vortices with the barotropic flow. Vortex interaction with a flow results in rotation variants, nutational oscillations, and unlimited stretching of its core. The vortex remains in a localized formation, with the semi-axes of the ellipse experiencing fluctuations near an average value in the first two cases. In the third case, the vortex is significantly elongated, and its shape in the horizontal plane changes as follows: one axis of the ellipse increases, and the other decreases. In this case, the vortex, when viewed from above, stretches into a thread, while remaining ellipsoidal. These vortex formations are called filaments. The latter arise from initially almost circular vortices in the horizontal plane and represent structures with non-zero vorticity elongated in one direction. Here, we aim to study the energy transformation of a vortex during its evolution process, mainly due to changes in its shape by stretching. The energy evolution of a mesoscale vortex located in the Norwegian Sea is analyzed using GLORYS12V1 ocean reanalysis data to verify the theoretical conclusions. During the evolution, the vortex is found to transform from a round shape and becomes elongated, and after three weeks its longitudinal scale becomes 4 times larger than the transverse one. During the transformation of a vortex, the kinetic energy and available potential energy decrease respectively by 3 times and 1.7 times. Concurrently, the total energy of the vortex is found to decrease by 2.3 times. We argue that the stretching of vortices results in a loss of energy as well as its redistribution from mesoscale to submesoscale. The lost part of the energy returns to the flow and results in the occurrence of the reverse energy cascade phenomenon.

**Keywords:** vortex; stretching; kinetic energy; available potential energy; GLORYS12V1; the Norwegian Sea

## 1. Introduction

Mesoscale vortices transfer heat, salt, mass, kinetic energy, and biogeochemical characteristics to thousands of km from the region of their formation. Although mesoscale vortices are formed almost everywhere in the world ocean except for the so-called "vortex deserts" [1], higher vortex activity is confined to areas of large-scale ocean currents. This is primarily due to the presence of baroclinic and barotropic instabilities of the current, which is the main reason for the generation of vortices. Nowadays, convincing evidence has been obtained that mesoscale vortices make a great contribution to the low-frequency variability in the sea level [1–5].

Geophysicists' interest in mesoscale variability in the world ocean is growing every year, but until recently, theoretical conclusions were ahead of empirical ideas. The availability of continuous satellite-derived sea level data and the development of new methodologies have revolutionized the studies focusing on the mesoscale variability in the ocean. Satellite altimeters allow you to measure the sea level with an accuracy of 1–2 cm relative to the reference Earth ellipsoid [6,7]. In recent years, many algorithms for tracking vortices using altimetric fields have been widely developed, which make it possible to establish various characteristics of vortices, such as size, polarity, western distribution, and lifetime. They showed that there is no steady connection between the polarities of vortices, their sizes, and propagation velocity [4].

With the development of computer technologies, it became possible to build global models of oceanic circulation and create various databases based on hydrodynamic modeling. For physical oceanographers, reanalysis products that are based on modeling with data assimilation are of great interest. Models assimilate altimeter data; other remote sensing observations such as sea surface salinity, in situ observations, as well as data from drifting buoys; ARGO buoy systems; etc. One of these products is GLORYS12V1 (Global Ocean Physics Reanalysis), which is used in this study.

Ocean mesoscale vortices are vortex currents with typical horizontal scales, usually exceeding the baroclinic radius of Rossby deformation, rotation speeds of 10–80 cm/s, and displacement speeds of 1–10 cm/s. According to their generation mechanism, mesoscale vortices in the ocean can be divided into four categories: (a) Frontal vortices (or rings), formed by cutting off meanders from large-scale jet currents; (b) Free vortices generated by baroclinic instability of currents; (c) Topographic vortices formed when flowing around the bottom relief; (d) Vortices induced by atmospheric influences. According to the sign of particle rotation, vortices are divided into cyclonic and anticyclonic.

Our work is devoted to a better understanding of the vortex cores' deformation, i.e., the so-called internal dynamics of vortices. Potential vorticity conservation for particles is valid for ocean vortices with horizontal scales of the order of the Rossby deformation radius or larger (e.g., see [8]). In other words, every particle that moves retains in time its potential vorticity. The potential vorticity conservation is equivalent to the conservation of the angular momentum of a liquid particle in a stratified rotating ocean [8]. If we set a stepwise distribution of the potential vorticity of particles outside and inside the core and make some reasonable simplifying assumptions, the problem of calculating the current function of the flow induced by the vortex will mathematically be equivalent to the problem of the gravitational potential of a body with the same shape as the vortex core [9]. Notably, unlike the gravitational potential problem at the core boundary, additional conditions associated with hydrodynamics must be met. These conditions, formulated mathematically, will lead to an integro-differential equation for estimating the evolution of the vortex core boundary. This approach known as the contour dynamics method in the 2D version allows us to numerically investigate the vortex core boundary behavior [10]. In the 3D version, it is called the surface dynamics method. Still, this problem is solved numerically because it is too complex for analytical approaches. However, if we choose an ellipsoid as the shape of the vortex core, and consider background flows that have a linear dependence on the flow velocity, then the integro-differential equation can be reduced to a set of ordinary differential equations for all parameters of the ellipsoid. In this case, the ellipsoid will remain an ellipsoid, but it can deform and rotate.

Field studies portrayed the simplest structure of a vortex, which consists of a core of a uniformly swirling liquid and the surrounding liquid. The swirling core induces movement not only within itself but also in the environment. Exchange across the core boundary is very difficult, so in the first approximation, it can be neglected. The boundary of the core is liquid and impermeable but can be easily deformed by the movements of the external or internal fluid. Higher potential vorticity gradients along the core boundary act as a barrier that prevents external particles from entering the core, and internal particles from leaving

it. Of course, this is true in the absence of turbulence, but in any case, the penetration of particles through the core boundary is very difficult.

The theoretical description of vortices with finite volume (area) was initiated by Kirchhoff [11], who showed that for plane hydrodynamics an equilateral elliptical region rotates uniformly without changing shape [12]. Further development in this direction is documented in many studies, in particular by Chaplygin (1948) who analyzed the deformation of the Kirchhoff vortex by a constant shear flow [13]. Later, Kida considered the dynamics of a Kirchhoff vortex in an inhomogeneous flow linearly dependent upon coordinates [14]. He showed that the vortex boundary has three behaviors: nutational oscillations, rotation, and unlimited stretching. Although the semi-axes of the ellipse experience fluctuations near an average value and remain limited at the same time, the vortex remains a localized formation in the first two cases. In the latter case, one of the axes increases indefinitely, and the second tends to zero. The vortex, while remaining elliptical, stretches into a vortex thread, i.e., filament. In the first two cases, while the vortex survives in the flow, it is destroyed by the flow in the latter case. Being filamentous, the vortex practically does not induce velocities and, in the presence of even very weak diffusion, it quickly dissipates.

Further detailed studies of the vortex behavior are presented in several papers [15–23]. Similar mathematical problems in the shallow water approximation were also touched upon in many studies [24–27]. Although the above approach is of great importance for theoretical two-dimensional hydrodynamics, it is still very far from describing a real ocean. For studying real ocean vortices, the effects of the rotation of the Earth, stratification of seawater, and three-dimensionality of the vortex core should also be considered. Note that the three-dimensional ellipsoidal vortices that account for rotation and stratification have been previously studied by V.V. Zhmur and co-authors [28–36]. The basic solution in [30], upon which all subsequent research on the topic is based, describes a uniformly swirling ellipsoidal vortex in a stationary ocean or a barotropic horizontally inhomogeneous flow. Zhmur and Pankratov [9,30] generalized the Kirchhoff and Kida vortices [11,12] to a three-dimensional case. It was found that if one of the semi-axes of an ellipsoidal vortex is vertical, then the qualitative behavior of such a vortex is similar to the behavior of a two-dimensional ellipsoidal vortex. If there is no background flow, then the vortex rotates uniformly without changing its shape. The rotation speed of the core boundary lags behind the rotation speed of the liquid particles and depends on the elongation of the core horizontally and its oblateness vertically. If the vertical size is parametrically increased to infinity, then the rotation speed of the core approaches the rotation speed of a two-dimensional elliptical Kirchhoff vortex. However, the vortex in the presence of a background flow is deformed and may be stretched into a filament. Moreover, the vortex behavior is very different in a background flow which is either barotropic or has a vertical shift. The stretching of a vortex in a horizontal plane decreases the fluid motion induced by it. The mode of unlimited vortex stretching results in the destruction of the vortex by the flow. In this paper, we study the evolution of a mesoscale vortex by stretching. It means that small-scale mixing across the vortex boundary is weaker than deformation at the mesovortex scale. At the same time, the decrease in energy when the vortex is stretched does not affect the mixing across the vortex boundary in any way. Our paper aims to study the vortex energy transformation by its stretching, as well as to verify the theoretical results with observations.

## 2. Theoretical Aspects

Many studies contributed to developing a theoretical approach for describing the behavior of intra-thermoclinic vortices with ellipsoidal core shape and semi-ellipsoidal near-surface vortices in equidistant barotropic flows [9,20–23,28–36]. The ocean is assumed to have a constant Väisälä–Brunt frequency. The Rossby number (Ro) is considered to be small:

$$\mathrm{Ro} = \frac{U}{|f|\,L} << 1.$$

In the above equation, $U$ is the characteristic horizontal velocity, whereas $L$ is the characteristic horizontal scale of the vortices, and $f$ is the Coriolis parameter. This expression is convenient if we consider the change in the horizontal scale as $L$, and the associated change in the characteristic horizontal velocity as $U$. It is this case that interests us.

Notably, there is also a different formulation of the Rossby number that does not use the characteristic horizontal dimension, $L$, but rather the vertical component of the rotor of the flow velocity (relative vorticity):

$$\mathrm{Ro} = \left| \frac{\mathrm{rot}_z \vec{u}}{f} \right|,$$

where $\mathrm{rot}_z \vec{u} = \frac{\partial v}{\partial x} - \frac{\partial u}{\partial y}$ is the vertical component of the rotor of the flow velocity (relative vorticity) $\vec{u} = \{u, v, w\}$ in the Cartesian coordinate system $(x, y, z)$.

In the initial theoretical formulation, the vortex core is a freely deformable ellipsoidal "water bag", filled with a liquid with a homogeneous potential vorticity 'σ'. Outside the vortex, the particles have zero potential vorticity in the entire region under consideration. In the quasi-geostrophic approximation at Ro << 1, the mathematical problem is reduced to a single equation for atmospheric pressure, through which all other hydrodynamic characteristics of fluid motion can be calculated. Similarly, in a more general formulation, the effect of equidistant currents on an ellipsoidal vortex was also considered. In particular, a vortex deforming under the action of currents could change its size, nevertheless remaining in the form of an ellipsoid. Details of these studies can be found in the works [9,20,21,30–32].

In this paper, we investigate the change in vortex energy at various levels of elongation in the horizontal direction of the vortex core. Let us start with the general energy expression in the quasi-geostrophic approximation [37]:

$$E = \frac{1}{2} \iiint \left[ \rho_\Sigma(x,y,z) \left( u^2(x,y,z) + v^2(x,y,z) \right) + \frac{g^2}{\rho_0(z)} \frac{\rho^2(x,y,z)}{N^2(x,y,z)} \right] dx\, dy\, dz, \quad (1)$$

where $\rho_0$ is the mean density of seawater, $v$ and $u$ are the meridional and zonal components of the flow velocity, $g$ is the acceleration due to gravity, $N$ is the Väisälä–Brunt frequency, and $\rho = (\rho_\Sigma - \rho_0)$ is the current density deviation to $\rho_\Sigma$ from $\rho_0$. The integration boundaries are determined by the vortex scales [38,39], where horizontal boundaries are determined by isolines of zero relative vorticity $\zeta = \frac{\partial v}{\partial x} - \frac{\partial u}{\partial y}$, and the integral is calculated from 0 to 1000 m depth. The first and second terms in Equation (1) are, respectively, the kinetic energy and the available potential energy of the vortex. If the integration is carried out over the entire space in which the vortex induces fluid motion or deforms isochoric surfaces, Equation (1) gives the total energy of the vortex, including the vortex core energy, as well as the external fluid energy captured in rotational motion. If the integration is carried out only over vortex core volume, then we exclude the energy of the external rotating fluid and, thus, only consider the energy of the vortex core. We consider the options.

The dimensionless parameter $\varepsilon$ is the degree of vortex elongation and is estimated as the ratio of its horizontal scales $\varepsilon = \frac{a}{b} \geq 1$, where $a$ and $b$ are the horizontal semi-axes of the ellipsoid core: $a$ is the semi-major axis, $b$ is the semi-minor axis, and $c$ is the vertical half-axis of the vortex. A dimensionless parameter of the vertical oblateness of the vortex core ($K$) is introduced as $K = \frac{N}{f} \frac{c}{r_0}$, where $r_0 = \sqrt{ab}$ is the effective radius of the vortex, and $N$ is the Väisälä–Brunt frequency averaged over the upper 1000 m depth. During the vortex deformation by a barotropic flow, the product of the vertical half-axes $a \times b$ and $r_0$, respectively, do not change. Consequently, $K$ is also preserved during the vortex deformation [9,30]. Notably, the latter is true only under the assumption of the Väisälä–Brunt frequency invariance. If it changes during the evolution of the vortex, then, accordingly, $K$ will also change.

Further, nontrivial transformations can modify the energy Equation (1) as a function of $\varepsilon$ and the compression parameter $K$. Below there are two variants of energy expressions identical to each other [40]:

$$E(\varepsilon, K) = \frac{2}{15}\pi\rho_0 r_o^3 c^2 \sigma^2 \frac{N}{f} \int_0^\infty \frac{d\mu}{\sqrt{(\mu^2 + \nu\mu + 1)(K^2 + \mu)}}, \tag{2}$$

$$E(\varepsilon, K) = \frac{3}{40\pi}\rho_0 \frac{V_0^2 \sigma^2}{c} K \int_0^\infty \frac{d\mu}{\sqrt{(\mu^2 + \nu\mu + 1)(K^2 + \mu)}}. \tag{3}$$

Here, $V_0 = \frac{4}{3}\pi abc$ is the volume of the vortex core, $\sigma$ is the potential vorticity inferred from Rossby [41], $\mu$ is the integration variable, and $\nu = \varepsilon + \frac{1}{\varepsilon} \geq 2$ is the horizontal vortex elongation, another dimensionless parameter. In a coordinate system with two horizontal axes $(x, y)$ and a vertical axis $z$, the potential vorticity ($\sigma$) is expressed in terms of the current function $\psi(x, y, z, t)$, $t$ is the time, and $f$ is the Coriolis parameter [41]:

$$\sigma = \Delta_h \psi(x, y, z, t) + \frac{\partial}{\partial z}\frac{f^2}{N^2}\frac{\partial\psi(x, y, z, t)}{\partial z}.$$

Note that in the above equation $\Delta_h \psi = \text{rot}_z \vec{u}$. In general, the Väisälä–Brunt frequency $N(z)$ depends on the vertical $z$ coordinate.

The more elongated the vortex, the greater the $\varepsilon$ and $\nu$. Elongated vortices result in high values of $\varepsilon$ and $\nu$ and, according to Equations (2) and (3), they will have lower energy. Equations (2) and (3) consider the total energy of the vortex, which includes the kinetic and available potential energy of the vortex core, as well as the energy of the external fluid trapped in motion by the vortex. The energy of the core will change when it is deformed. Zhmur and Harutyunyan [40] analytically calculated the kinetic $H_{core}^k$ and available potential energy $H_{core}^p$ of the vortex core as functions of the parameters $\varepsilon$ and $K$ for ellipsoidal vortices:

$$E_{core}^k(\varepsilon, K) = \frac{1}{40}\rho_0\sigma^2 VabK^2 \left( \varepsilon\left[\int_0^\infty \frac{ds}{\sqrt{(\varepsilon+s)^3(\varepsilon^{-1}+s)(K^2+s)}}\right]^2 + \varepsilon^{-1}\left[\int_0^\infty \frac{ds}{\sqrt{(\varepsilon+s)(\varepsilon^{-1}+s)^3(K^2+s)}}\right]^2 \right), \tag{4}$$

$$E_{core}^p(\varepsilon, K) = \frac{1}{40}\rho_0\sigma^2 VabK^4 \left[\int_0^\infty \frac{ds}{\sqrt{(\varepsilon+s)(\varepsilon^{-1}+s)(K^2+s)^3}}\right]^2, \tag{5}$$

where $s$ is the integration variable. In this case, the total mechanical energy of the vortex core $E_{core}(\varepsilon, K)$ is the sum of $E_{core}^k(\varepsilon, K)$ and $E_{core}^p(\varepsilon, K)$:

$$E_{core}(\varepsilon, K) = E_{core}^k(\varepsilon, K) + E_{core}^p(\varepsilon, K). \tag{6}$$

Theoretically, when a vortex is elongated by a barotropic flow, only $\nu$ and $\varepsilon$ change in Equations (2) and (3) [40]. The denominator in the integrand increases with the elongation of the vortex, and with elongation the integral itself decreases. In other terms, during vortex stretching, kinetic and available potential energy, as well as the total mechanical energy of the vortex core, decrease. The maximum energy values in Equations (2) and (3) at a fixed $K$ correspond to round vortices in a horizontal plane with $\varepsilon = 1$ or $\nu = 2$. At the same time, when the background frequency of the Väisälä–Brunt frequency changes, the parameter $K$ will also change.

Next, we will investigate the transformation of the kinetic and available potential energy of a quasi-permanent mesoscale vortex using the Global Ocean Physics Reanalysis

and compare it with the theoretical estimates. The analyzed vortex is located in the Norwegian Sea.

## 3. Data and Methods

The GLORYS12V1 (Global Ocean Physics Reanalysis) data, a global ocean vortex-resolving reanalysis with a spatial resolution of $1/12°$ at 50 levels is available via the CMS (Copernicus Marine Service). GLORYS12V1 is based on the CMS global real-time forecasting system, where the NEMO model with ECMWF ERA-Interim forcing is used to simulate oceanic conditions. In situ measurements, data from altimeters (sea level anomaly), sea surface temperature (SST), sea ice cohesion, and vertical temperature and salinity profiles are assimilated together. Observations are assimilated using a low-order Kalman filter. The product includes daily 3D fields of potential temperature, salinity, and currents, as well as 2D fields of sea level, mixed layer depth, potential bottom temperature, ice thickness, ice types, and ice drift velocities.

To estimate the vortex area and to exclude fragments of other hydrodynamic structures in the MATLAB environment, we constructed masks for each day during the vortex transformation by stretching [42]. At the first stage, in order to calculate the geometric parameters of the vortex during the process of its transformation into a filament, it is necessary to bring its horizontal scales to a single spatial discreteness since MATLAB considers a single segment (pixel) as the $x$ and $y$ steps. Using the *kron* function, we obtain the values of one step in $x$ equal to 0.3165 km and in $y$ equal to 0.3193 km. Thus, the length of one pixel in the calculations is ~0.318 km. After the $x$ and $y$ steps have been aligned, various geometric characteristics of the vortex can be calculated: the large and small semi-axes of the vortex, their orientation, perimeter, area, etc., as well as the vertical oblateness parameter.

## 4. Analysis of Changes in the Kinematic Characteristics of a Mesoscale Vortex in the Process of Its Stretching

Figure 1 shows an example of the lifespan of an anticyclonic vortex in the Norwegian Sea. The evolution of the vortex during the process of its stretching is monitored continuously for 22 days (3–24 April 2012). The vortex, which started as a round structure (3 April 2012; Figure 1a), is gradually seen to stretch and elongate with time elapse (Figure 1b–e). After nearly 2 weeks (19 April 2012, Figure 1d), the semi-major axis of the vortex is several times larger than the semi-minor axis. By April 21 the vortex is further elongated in the longitudinal direction, and by April 24, the vortex bending under the influence of the flow attains a horseshoe shape.

Figure 2 shows the characteristics of the vortex: the elongation parameter, $\varepsilon$; and the effective radius $r_0 = \sqrt{ab}$. The vortex is round ($\varepsilon = 1$) in the horizontal plane at the start of its lifespan. During its lifetime, the vortex gradually elongates, so that the longitudinal scale of the vortex is much larger (4 times) than the transverse one. The effective radius decreased during the initial 12 days but increased thereafter. Note that the effective radius of the vortex at the start and the end of the vortex lifespan are not very different in magnitude.

Figure 3 shows the evolution of the Väisälä–Brunt frequency and the dimensionless parameter, $K$. Following Zhmur et al. [43], the vertical semi-axis is assumed to be equal to 400 m in calculations. Vortex characteristics, $K$ and $N$, both increase during the vortex lifespan. The increase in the Väisälä–Brunt frequency ($N$) may be associated with the period selected for the analysis. Note that April is the period after the winter convection during which the stratification gradually strengthens [44–46]. The influence of winter convection likely persists over the vortex during the start of the time series, while the subsequent increase in water stratification during the following days may increase $N$. The increase in the frequency of the Väisälä–Brunt frequency $N$ is likely to increase the oblateness parameter $K$, as also seen from the figure. We can obtain the estimation of $K$.

$$K = \frac{N}{f} \frac{3 V_0}{4 \pi r_0^3} \tag{7}$$

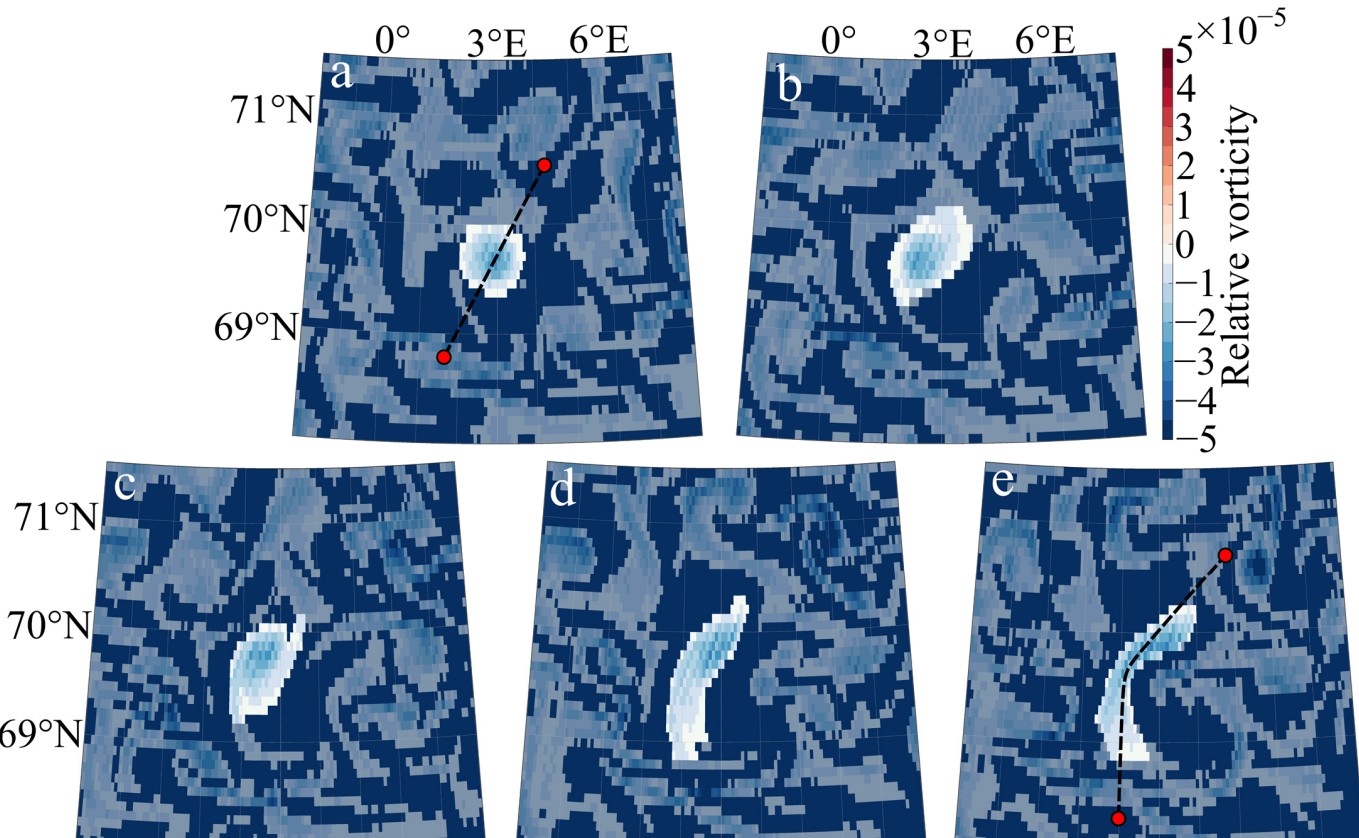

**Figure 1.** Vortex evolution during the period of stretching: (**a**) 3 April 2012, (**b**) 9 April 2012, (**c**) 14 April 2012, (**d**) 19 April 2012, (**e**) 24 April 2012. The scale shows the relative vorticity values; the horizon is 541 m. The dashed lines and red circles in panels (**a**) and (**e**) are used later.

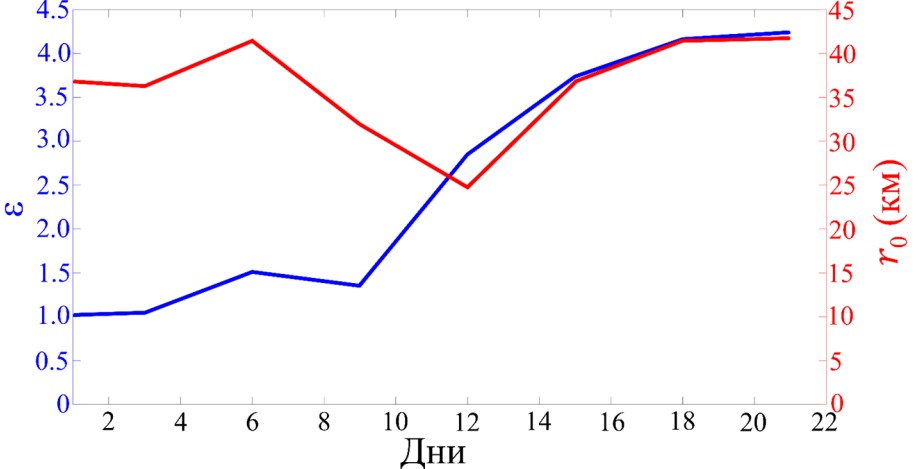

**Figure 2.** Dimensionless parameter of the vortex horizontal elongation $\varepsilon$ (blue curve) and its effective radius $r_0 = \sqrt{ab}$ (red curve). The x-axis shows the days of the vortex lifecycle from the beginning of measurements on 3–24 April 2012.

Here, c is the size of the vertical semi-axis, $V_0$ is the volume of the vortex core, and $r_0 = \sqrt{a\,b}$.

According to Figure 2, the effective radius $r_0$ changes during the vortex deformation, but its initial and final values are almost the same. Similarly, it is also safe to assume that the volume of the vortex core also does not change. As a result, the only parameter that can change K is the Väisälä–Brunt frequency N. Figure 3 demonstrates that the increase in

parameter K is in phase with the increase in the Väisälä–Brunt frequency N. This gives the confidence to conclude that the growth of K is solely due to an increase in N.

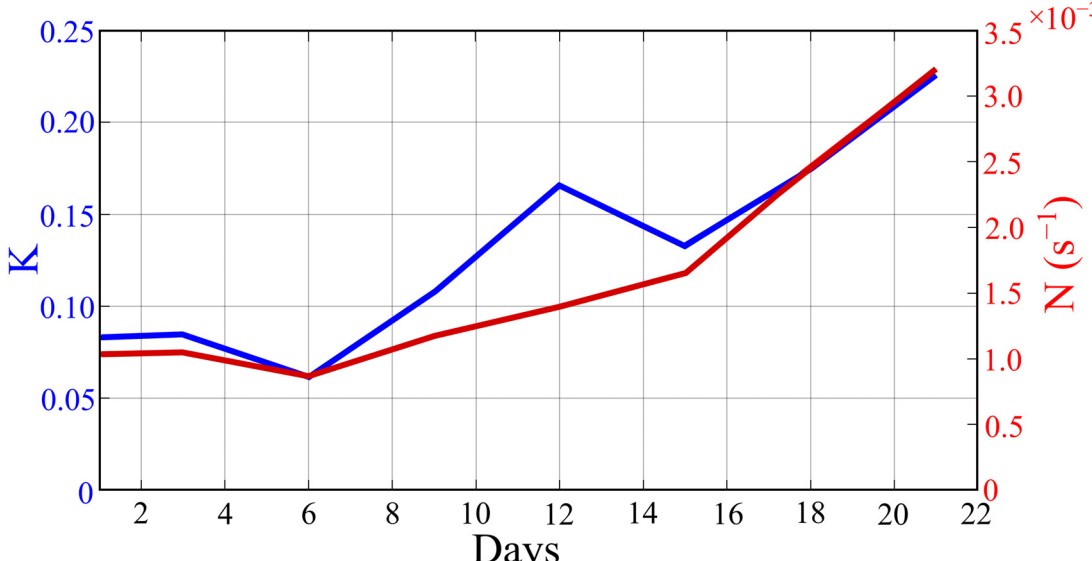

**Figure 3.** Dimensionless parameter of the vortex core vertical oblateness K (blue curve) and the Väisälä–Brunt frequency N (red curve). The abscissa axis shows the days of the vortex lifecycle from the beginning of measurements on 3–24 April 2012.

Next, we analyze separately the estimates of kinetic and available potential energy terms in Equation (1). The horizontal integration area is bound by the vortex area (see Figure 1), and the vertical integral is taken from 0 to 1000 m depth [5,38,39]. Analysis (Figure 4) quantifies that the available potential energy of the vortex is 1.5 times higher than its kinetic energy. During the vortex lifespan, the kinetic and the potential energy is reduced to 3 and 1.7 times, respectively. This decrease in energy is associated with a change in the shape of the vortex and its stretching. The lower rate of potential energy decrease, in comparison to the rate of kinetic energy, is related to the Väisälä–Brunt frequency, an increase in which during the end of the vortex lifespan slows down the decrease in the available potential energy of the vortex.

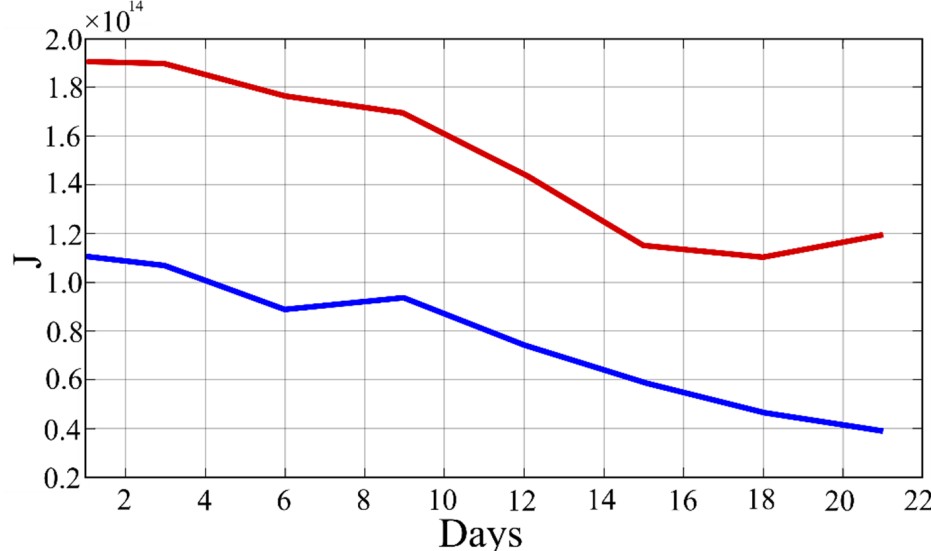

**Figure 4.** Kinetic (blue curve) and available potential (red curve) energy of the vortex (J). The x-axis shows the days of the vortex lifecycle from the beginning of measurements on 3–24 April 2012.

The change in the available potential and kinetic energy of the vortex associated with its horizontal elongation (the $\varepsilon$ parameter) is analyzed next. Analysis (Figure 5) shows a linear decrease in energy with an increase in the elongation parameter. According to Figures 4 and 5, the total energy of the vortex when it is stretched decreases by about 2.3 times. The vortex core transformation during its elongation is also confirmed by the changes in the thermohaline characteristics of the vortex. Figure 6 shows the temperature longitudinal cross-sections of the vortex. When the vortex is stretched, the part of the core limited to the 5° C isotherm is compressed several times. The vortex still had a round shape at the start (3 April 2012), when the 5 °C isotherm is located at 600 m depth, but retreated to ~300 m depth by 21 April. In other terms, the area is reduced in depth by half during the lifetime of the vortex. On the contrary, the part of the core bounded by the 4.5 °C isotherm stretches along the vortex in the longitudinal direction. Note that in Figure 6, the vortex is not an ellipsoid but rather a semi-ellipsoid. However, there is no contradiction in this since the theory of ellipsoidal vortices can be also applied to cases of subsurface vortices when a semi-ellipsoid is considered as a vortex [43]. Similar changes in salinity and potential density isolines are also found during the analysis.

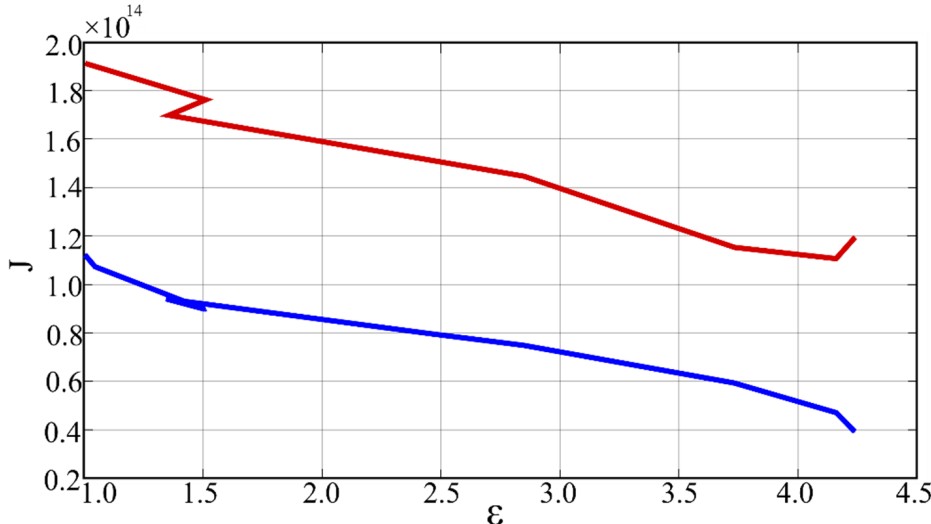

**Figure 5.** Dependence of kinetic (blue curve) and potential (red curve) energy on the dimensionless parameter of horizontal elongation $\varepsilon$ inferred from the in situ data.

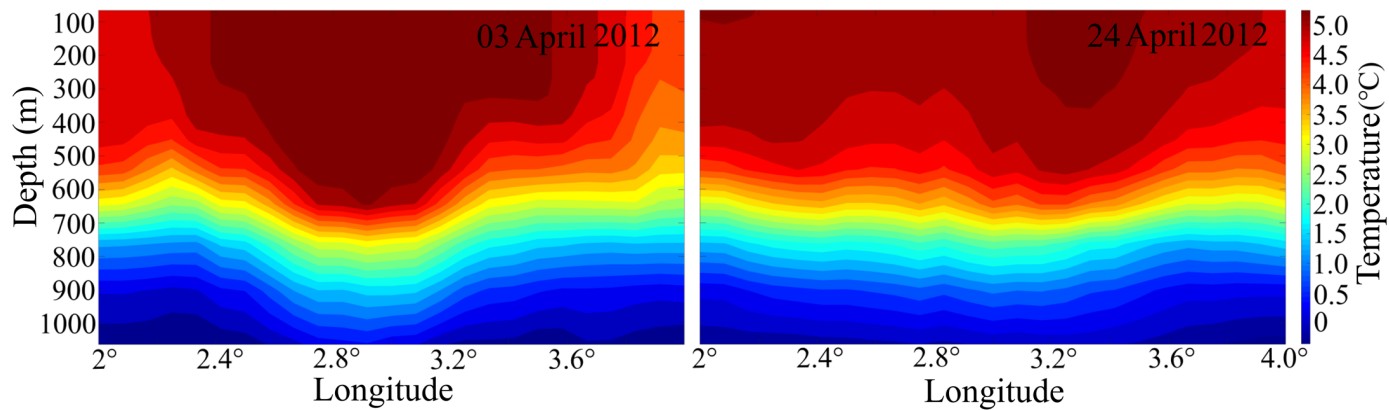

**Figure 6.** The vertical cross-section of temperature (°C) in the vortex for 3 and 24 April 2012.

Thus, along with a decrease in kinetic and available potential energy in the vortex, a core transformation occurs, in which the area with the maximum value of temperature, salinity, and potential density decreases in size but stretches with lower temperature as well as salinity and potential density values, in the longitudinal direction of the vortex.

The methodology of calculating energy according to oceanic reanalysis data, given above, is similar to analytical calculations in Equations (4)–(6), which makes it possible to compare theoretical estimates with those based on reanalysis data. The sequential evolution of the vortex, according to Figure 3, starts (3 April 2012) at $\varepsilon = 1$ and K = 0.08 and ends (24 April 2012) at $\varepsilon = 4.3$ and K = 0.23. Using Equations (2)–(5), we can calculate the energy ratio $E_{core}^{k}(\varepsilon, K)$, $E_{core}^{p}(\varepsilon, K)$, $E_{core}(\varepsilon, K)$ at the final stage to their initial value:

$\dfrac{E_{core}^{k}(1;0.08)}{E_{core}^{k}(4.3;0.23)}$ —the total kinetic energy of the vortex core,

$\dfrac{E_{core}^{p}(1;0.08)}{E_{core}^{p}(4.3;0.23)}$ —the total available potential energy of the vortex core,

$\dfrac{E_{core}(1;0.08)}{E_{core}(4.3;0.23)}$ —the total energy of the vortex core.

We will interpret these relations as the coefficient of attenuation of the corresponding types of energy. Moreover, this can be performed as in the theoretical approach according to Equations (4)–(6), as with the reanalysis data. A similar attenuation coefficient can be obtained for the total energy of the vortex $E(\varepsilon, K)$, which additionally includes the energy of the rotating fluid external to the core: $\dfrac{E(1;0.08)}{E(4.3;0.23)}$. However, the latter can be conducted only on the basis of theoretical relations in Equations (2) and (3), since such a calculation is impossible using reanalysis data. Table 1 shows all the specified attenuation coefficients.

**Table 1.** The attenuation coefficients of various types of energy during the vortex stretching.

| $\dfrac{E(1;0.08)}{E(4.3;0.23)}$ | $\dfrac{E_{core}(1;0.08)}{E_{core}(4.3;0.23)}$ | $\dfrac{E_{core}^{k}(1;0.08)}{E_{core}^{k}(4.3;0.23)}$ | $\dfrac{E_{core}^{p}(1;0.08)}{E_{core}^{p}(4.3;0.23)}$ |
|---|---|---|---|
| Based on the GLORYS12V1 data | | | |
| - | 0.43 | 0.33 | 0.59 |
| Based on theory | | | |
| 0.80 | 0.53 | 0.53 | 0.54 |

According to Table 1, when the vortex is stretched, there is a decrease in all types of energy: kinetic, available potential, and total mechanical energy of the vortex core. We observe not only qualitative but also acceptable quantitative correspondence of theoretical and practical assessments. Unfortunately, on the basis of reanalysis, it is not technically possible to estimate the total energy of the vortex, which includes not only the vortex core energy but also the energy of the liquid trapped outside the core. Therefore, there are no corresponding estimates in Table 1. The bottom row of Table 1 shows the results of theoretical calculations using Equations (2)–(5). When the vortex is stretched, the core energy is found to decrease faster than the total energy of the vortex, which includes the energy of the external rotating water. This means that the energy of the external fluid decreases less intensively when the vortex is stretched, in comparison to the decrease in the total energy of the vortex and its core energy. Unfortunately, this conclusion based on theory cannot be confirmed experimentally by our methods. However, since the rest of the theoretical and observed values correspond to each other, it is expected that the conclusion will be fair too. Discussion of the energy external to the core of the vortex region is appropriate, since this part of the vortex has, according to theory, noticeably more energy than the core itself [40].

## 5. Interpretation of the Results and Discussion

We found out that when the vortices are stretched out their energy decreases. Vortices are stretched by an inhomogeneous current, which at the same time performs work on the vortex. This work is spent on reducing the energy of vortices. If we consider an ensemble of vortices of different signs, sizes, and intensities against the background of an inhomogeneous barotropic flow, the behavior of such an ensemble will be qualitative as follows. At first, closely located vortices of the same sign will merge and form larger vortices of the same sign [9]. Closely spaced vortices of different signs form dipole pairs [47].

If the integral potential vorticity of such pairs turns out to be zero, then the pairs will leave the territory of the ensemble. If the integral vortices of the pairs are not compensated, then the center of mass of such pairs will move along a large circle, periodically leaving and returning to the ensemble zone. The remaining vortices will interact with each other more or less actively. A more active interaction will be reduced to stretch weak vortices by strong ones. All this will be accompanied by the stretching effect of the flow on each vortex and the ensemble as a whole. As a result, after the initial period of merging close vortices of the same name, it is natural to expect the following situation. Some of the sufficiently strong vortices will survive in the flow and will evolve without significant stretching, i.e., they will remain localized formations. Their evolution will mainly be reduced to the movement of vortices by the flow and the interaction of vortices with each other. The remaining relatively weak vortices, or not weak yet but already elongated initially, will continue to be stretched by the flow or by other vortices [9]. The energy of the surviving vortices will not change on average, but the energy of the stretching vortices will decrease. In total, the energy of the ensemble will decrease due to a decrease in the energy of the elongating vortices. If we approach this problem from the viewpoint of turbulence theory, where vortices play the role of turbulence elements, then in such a system the total turbulence energy will be lost over time. These losses will return to the average flow since in our speculative model there is only flow and turbulence and nothing else. Such a physical behavior of an ensemble of mesoscale vortices leads to two qualitative conclusions: 1. As the vortices elongate, they decrease in transverse dimensions, which corresponds to the redistribution of energy from the mesoscale dimensions toward the submesoscale. This is a direct energy cascade. 2. Since not all the energy is transferred to a longer vortex during its elongation, the "lost" energy is returned to the background flow at a characteristic horizontal scale exceeding the submesoscale. This is a property of the reverse energy cascade, i.e., the redistribution of energy from small to large scales. Initially, this phenomenon was called "the negative viscosity phenomenon" [48]. With the help of a direct energy cascade, energy is redistributed continuously from the mesoscale to the submesoscale (transverse scale of vortices), while the reverse energy cascade "transfers" energy from small scales to the flow scale without "stopping" at intermediate scales (as a tunnel effect). This means that at sizes from the mesoscale to the submesoscale, these oppositely directed energy flows do not intersect with each other and do not compensate for each other. The existence of a reverse energy cascade means that in regions where vortices can be stretched by the flow, the negative viscosity properties should be expected to manifest.

The general picture of the evolution of an ensemble of quasi-geostrophic vortices against the background of a large-scale flow is as follows. Mainly as a result of baroclinic instability, the flow itself generates mesoscale vortices, some of the vortices survive the flow, and some are stretched out by the flow and transformed into submesoscale vortex formations (filaments). Stretching elements lose energy. Due to the loss of energy in stretching vortices and the return of energy back into the flow, the phenomenon of a reverse energy cascade or "the negative viscosity phenomenon" occurs.

Thus, as a result of the vortex stretching, the energy is transferred to other, smaller scales, i.e., the effect we have indicated leads to a redistribution of energy from mesoscales to the submesoscale [34,35]. According to this approach, energy is transferred from scale to scale with losses. The word "loss" in this case implies the part of the energy that is returned to large-scale flows, and the rest of the energy is redistributed to the submesoscale. The interaction of the individual vortex with the background current leads to the stretching of the vortex and, as a consequence, the loss of energy by the vortex, which passes into the current. In the case of an ensemble of vortices, at the initial stage of interaction, their merging is possible and, as a consequence, the enlargement of the newly formed vortex occurs. For a barotropic ocean, this is an essential factor, since the critical distance between the centers for the subsequent merger of vortices is quite large, i.e., 3.2 R, where R is the radius of the initial vortices [49]. However, baroclinic vortices can merge at much smaller distances between the centers [29]. As a result, the efficiency of the merging of

baroclinic vortices will be lower than for barotropic vortices; therefore, the enlargement of baroclinic vortices due to merging is less likely. On the contrary, the stretching of vortices into filaments is much easier for baroclinic vortices and, therefore, it should be expected that this process will become prevalent. During the formation of vortices from the flow due to baroclinic instability, e.g., the energy from the currents is transferred to the characteristic scales of the vortices (direct energy cascade). When the vortices are stretched into filaments, the process of energy redistribution to smaller scales occurs (this is also a direct energy cascade). However, the process of stretching vortices is accompanied by a loss of energy, which returns to the current (inverse energy cascade).

The presence of the directions in which the vortices are stretching can lead to the anisotropy of the medium, and in turn, this can lead to a deviation from Kolmogorov's theory. However, these effects have been poorly studied and there is currently no confidence in their significance. When the shape of the vortex is lengthened, a highlighted direction of its elongation appears, which means a violation of the isotropy of space. The hypothesis of local isotropy of turbulence may also be violated. Unlimited stretching of vortices does not occur in any inhomogeneous flows but only in flows with pronounced deformation properties that satisfy certain conditions imposed on the flow parameters. These conditions are discussed in detail in the previous studies [34–36]. Obviously, the flow in which the mesoscale vortex is located (shown in Figure 1) satisfies these properties.

## 6. Conclusions

We analyzed the mesoscale vortex energy transformation that changes its shape during the process of evolution by stretching. Our analysis (Equations (2)–(3)) found that during such a transformation, there is a decrease in the kinetic and available potential energy of the vortex. It is also found that the horizontal semi-axes and the effective radius change slightly when the vortex is deformed by a barotropic flow (see Figure 2). The vortex energy change during its transformation is analyzed based on the parameter $\varepsilon$, which characterizes the ratio of its horizontal axes, and the vertical oblateness parameter of the vortex core $K$. The kinetic and available potential energy enclosed in the volume of the vortex core (i.e., the total mechanical energy of the vortex) are considered as functions of parameters $(\varepsilon, K)$. A decrease in energy with an increase in $\varepsilon$ has been experimentally proven (see Figure 5).

Theoretical conclusions are verified by analyzing the energy evolution of a mesoscale vortex located in the Norwegian Sea. The study was conducted using GLORYS12V1 ocean reanalysis data. During the 22 days of evolution (3–24 April 2012), the vortex which had a round shape at the start was stretched, such that the longitudinal scale of the vortex was found to be much larger (4 times) than the transverse one. Note that the change in effective radius during the evolution was minimal, as its values at the start and at the end of the vortex lifecycle are comparable in magnitude. It is also established that the increase in the vortex parameter $K$ is associated with an increase in the Väisälä–Brunt frequency $N$.

The analysis found that the available potential energy of the vortex is higher (1.5 times) than its kinetic energy (Figure 4). The energy loss during vortex transformation occurs in different ways such as a decrease in the kinetic energy (3 times) and the available potential energy (1.7 times). In concurrence, there is a decrease in the total vortex energy (2.3 times). This decrease in energy is due to the stretching of the vortex and the associated change in the vortex shape. A linear decrease in the available potential and kinetic energy of the vortex based on the elongation parameter $\varepsilon$ is found (Figure 5). Inferred from ocean reanalysis data, the estimates of the coefficients of relative attenuation of various vortex energy types during vortex evolution by stretching qualitatively confirm the theoretical estimates. Note that the lack of complete correspondence between quantitative estimates may be associated with the inaccuracy of the determination of the vortex scale from reanalysis data, as well as some uncertainties.

As the vortex core is pulled out by the barotropic flow, the geometric parameters of the vortex change as follows. One horizontal axis increases indefinitely, and the second one decreases while saving the product of $a \times b$. As a result, the core becomes like a long

"ribbon" stretched horizontally with a finite vertical size equal to the initial vertical size of the vortex core. From above, such a "ribbon" looks like a vortex thread or filament.

The stretching of vortices leads to a redistribution of energy from the mesoscale to the submesoscale with a loss of energy. The lost part of the energy is returned to the flow, which causes the phenomenon of the reverse energy cascade or "negative viscosity phenomenon". The stretching of vortices by currents can cause deviations from the law of $-5/3$ in the spatial energy spectrum.

**Author Contributions:** Conceptualization, V.V.Z. and T.V.B.; methodology, V.V.Z.; software, V.S.T.; validation, T.V.B., V.S.T. and E.V.N.; formal analysis, D.A.H.; investigation, T.V.B.; resources, V.S.T.; data curation, V.S.T.; writing—original draft preparation, T.V.B.; writing—review and editing, R.P.R.; visualization, V.S.T.; supervision, T.V.B.; project administration, T.V.B.; funding acquisition, T.V.B. All authors have read and agreed to the published version of the manuscript.

**Funding:** This research was funded by Russian Science Foundation, grant number 22-27-00004, St. Petersburg State University, grant number 94033410, and Shirshov Institute of Oceanology, Russian Academy of Sciences, State Contract No. 0128-2021-0002. The APC was funded by St. Petersburg State University.

**Institutional Review Board Statement:** Not applicable for studies not involving humans or animals.

**Informed Consent Statement:** Not applicable for studies not involving humans.

**Data Availability Statement:** We used the GLORYS12V1 (Global Ocean Physics Reanalysis) data, a global ocean vortex-resolving reanalysis with a spatial resolution of $1/12°$ at 50 levels is available via the CMS (Copernicus Marine Service): https://data.marine.copernicus.eu/product/GLOBAL_MULTIYEAR_PHY_001_030/description.

**Acknowledgments:** The publication was funded by the Russian Science Foundation (project No. 22-27-00004), St. Petersburg State University (project No. 94033410), and State Contract of Shirshov Institute of Oceanology, Russian Academy of Sciences No. 0128-2021-0002.

**Conflicts of Interest:** The authors declare no conflict of interest.

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
