# Peer review of "Changes in the Available Potential and Kinetic Energy of Mesoscale Vortices When They Are Stretched into Filaments"

_jmse, doi:10.3390/jmse11061131_

Round 1
Reviewer 1 Report
Comments and Suggestions for Authors
This manuscript presents theoretical arguments regarding the transformation of energy in a mesoscale barotropic vortex due to horizontal deformation / stretching and applies it to an example mesoscale eddy in the Norwegian Sea, using GLORYS12V1 ocean reanalysis data to verify the theoretical conclusions. The degree of vortex elongation is used to quantify the change in shape horizontally. Since potential vorticity is used as a primary variable, changes in the vertical mass distribution of the vortex, including changes in static stability, can be assessed. A stepwise increase in PV is assumed across the eddy boundary, with the flow associated with the PV anomaly calculated by a balance relation. A dimensionless parameter of the vertical oblateness of the vortex core relative to the horizontal scale is introduced, which quantifies the departure from Prandtl’s ratio scaling. By introducing these parameters into a conservation relation for eddy kinetic energy, two energy variant expressions were derived and shown as (2) and (3). It seems important that this step should be described further, so that the reader can understand it better. (For example, it is hard to guess what the parameter mu is, perhaps a variable of integration related to wavenumber interval?) Similarly, in (4) and (5), what is s? It would help the reader to explain the physical significance and interpretation of these equations a little bit more. Taking these somewhat on faith, I was quite interested in application to the results. The quantification of energy loss by the eddy with this method seems to me to be a new contribution. The figures are interesting. I recommend publication with minor revision. The authors may wish to consider the points below, in order of encountering the ideas in the paper.
1. par. 6, “ Exchange across the core boundary is very difficult, so in the first approximation, it can be neglected.” Is this a statement that small-scale mixing across the boundary is weaker than deformation at the mesovortex scale? One could invoke two transport time scales, one at the eddy-advection scale and one at the small-scale mixing scale.
2. A primary conclusion is that there is loss of energy by the eddy due to stretching, which is interesting and has important implications for transport. e.g., in a tokomak one of the main ways of trying to suppress eddies is to shear them. This seems to be an important aspect of containment of the stratospheric polar vortex for radial mixing of constituents, that there is strong shear on the outside of the vortex speed maximum which suppresses eddies. Does a decline in vortex energy imply more rapid mixing in this case?
3. You have several comments about the energy cascade. If this eddy lost energy, were there ones nearby that might have become stronger as a result? Was a larger-scale eddy created? Is it possible to relate your ideas to departures from Kolmogorov theory, where some eddies live longer, such as salty Mediterranean eddies, or “Meddies”, yielding a local spike in the spectral profile and an “eddy gerontocracy”? (In the last two paragraphs of section 5, maybe you are saying that energy goes both up and downscale, not just downscale?)
4. Just after defining Ro: The “rotor flow velocity” is usually referred to as the relative vorticity in atmospheric sciences. I see that after this you use relative vorticity. I guess it’s okay to say it different ways. Maybe I am missing something and they are different?
5. Eqns. (2)-(5): Please explain what mu is - wavenumber? How are they derived? Please give a citation or more detail of this derivation.
6. Fig. 1: Please explain the color bar further in the caption. It looks like there are no positive regions shown, or is everything with value larger than zero colored dark blue? Please state that the dashed lines in panels a and e are used in Fig. 6a, b.
7. This paper “established that the increase in the vortex parameter K is associated with an increase in the Väisälä–Brunt frequency N”, which seems useful.
Author Response
Dear reviewer,
Thank you very much for your consideration, and we really appreciate the comments. We are very grateful to you for your comments which undoubtedly helped to improve our manuscript.
Appropriate changes were made and highlighted in yellow in the revised manuscript according to the suggestions of the reviewers.
This manuscript presents theoretical arguments regarding the transformation of energy in a mesoscale barotropic vortex due to horizontal deformation / stretching and applies it to an example mesoscale eddy in the Norwegian Sea, using GLORYS12V1 ocean reanalysis data to verify the theoretical conclusions. The degree of vortex elongation is used to quantify the change in shape horizontally. Since potential vorticity is used as a primary variable, changes in the vertical mass distribution of the vortex, including changes in static stability, can be assessed. A stepwise increase in PV is assumed across the eddy boundary, with the flow associated with the PV anomaly calculated by a balance relation. A dimensionless parameter of the vertical oblateness of the vortex core relative to the horizontal scale is introduced, which quantifies the departure from Prandtl’s ratio scaling. By introducing these parameters into a conservation relation for eddy kinetic energy, two energy variant expressions were derived and shown as (2) and (3). It seems important that this step should be described further, so that the reader can understand it better. (For example, it is hard to guess what the parameter mu is, perhaps a variable of integration related to wavenumber interval?) Similarly, in (4) and (5), what is s?
Thank you! We added the missing reference and explained µ and s.
It would help the reader to explain the physical significance and interpretation of these equations a little bit more. Taking these somewhat on faith, I was quite interested in application to the results. The quantification of energy loss by the eddy with this method seems to me to be a new contribution. The figures are interesting. I recommend publication with minor revision. The authors may wish to consider the points below, in order of encountering the ideas in the paper.
The authors sincerely thank you for the high appreciation of our work.
- par. 6, “ Exchange across the core boundary is very difficult, so in the first approximation, it can be neglected.” Is this a statement that small-scale mixing across the boundary is weaker than deformation at the mesovortex scale? One could invoke two transport time scales, one at the eddy-advection scale and one at the small-scale mixing scale.
Clarified.
- A primary conclusion is that there is loss of energy by the eddy due to stretching, which is interesting and has important implications for transport. e.g., in a tokomak one of the main ways of trying to suppress eddies is to shear them. This seems to be an important aspect of containment of the stratospheric polar vortex for radial mixing of constituents, that there is strong shear on the outside of the vortex speed maximum which suppresses eddies. Does a decline in vortex energy imply more rapid mixing in this case?
Clarified (Page 3).
- You have several comments about the energy cascade. If this eddy lost energy, were there ones nearby that might have become stronger as a result? Was a larger-scale eddy created? Is it possible to relate your ideas to departures from Kolmogorov theory, where some eddies live longer, such as salty Mediterranean eddies, or “Meddies”, yielding a local spike in the spectral profile and an “eddy gerontocracy”? (In the last two paragraphs of section 5, maybe you are saying that energy goes both up and downscale, not just downscale?)
We added two paragraphs to Section 5 and also two appropriate references.
- Just after defining Ro: The “rotor flow velocity” is usually referred to as the relative vorticity in atmospheric sciences. I see that after this you use relative vorticity. I guess it’s okay to say it different ways. Maybe I am missing something and they are different?
Thank you! Corrected.
- Eqns. (2)-(5): Please explain what mu is - wavenumber? How are they derived? Please give a citation or more detail of this derivation.
The integration variable µ is explained. The derivation of equations (2)-(5) is given in the published article (Zhmur and Harutyunyan, 2023). The reference is given.
- Fig. 1: Please explain the color bar further in the caption. It looks like there are no positive regions shown, or is everything with value larger than zero colored dark blue? Please state that the dashed lines in panels a and e are used in Fig. 6a, b.
We used the masks for each day during the vortex transformation by stretching (the description is on Page 6). This allows us to use MATLAB to exclude fragments of other hydrodynamic structures. Many thanks for the comment about the dashed lines. We added this to the caption of Fig. 1.
- This paper “established that the increase in the vortex parameter K is associated with an increase in the Väisälä–Brunt frequency N”, which seems useful.
Thank you!

Reviewer 2 Report
Comments and Suggestions for Authors
This paper discusses how the energy within eddies change when they are stretched into filaments. They present both analytical solutions and diagnose a realistic eddy using GLORYS dataset. The topic is interesting; however, the paper needs to be improved in several aspects.
(1) I strongly recommend the authors edit their manuscript. Right now, the English writing is of relatively poor quality, which makes it a bit challenging to read through.
(2) Please explain how one derive Equations (1)-(3). More details need to be provided.
(3) Only one vortex is analyzed using the GLORYS data. Since eddies are prevalent in the ocean, I recommend you to track all the vortices in a specific region and then provide a statistical description of the energy change during eddy evolution.
(4) The theory is tailored to barotropic flow. However, in the realistic scenario, both barotropic and baroclinic flows exist. Please explain the applicability of this theory to realistic cases.
Comments on the Quality of English LanguageI strongly recommend the authors edit their manuscript. Right now, the English writing is of relatively poor quality, which makes it a bit challenging to read through.
Author Response
Dear reviewer,
Thanks so much for sharing your experience with us. We are very grateful to you for your comments which undoubtedly helped to improve our manuscript.
Appropriate changes were made and highlighted in yellow in the revised manuscript according to the suggestions of the reviewers.
Comments and Suggestions for Authors
This paper discusses how the energy within eddies change when they are stretched into filaments. They present both analytical solutions and diagnose a realistic eddy using GLORYS dataset. The topic is interesting; however, the paper needs to be improved in several aspects.
- I strongly recommend the authors edit their manuscript. Right now, the English writing is of relatively poor quality, which makes it a bit challenging to read through.
Thank you. We checked the text carefully using the available services and tried to do our best to improve the language in the revised manuscript.
- Please explain how one derive Equations (1)-(3). More details need to be provided.
The derivation of equations (2)-(5) is given in the published article (Zhmur and Harutyunyan, 2023). The reference is given.
- Only one vortex is analyzed using the GLORYS data. Since eddies are prevalent in the ocean, I recommend you to track all the vortices in a specific region and then provide a statistical description of the energy change during eddy evolution.
This is our first research about the transformation of the energy of an individual vortex when it is stretched. However, we published several papers concerning the spatial distribution of areas in the World Ocean where vortex stretching into filaments is possible and where vortex stretching is prohibited. In these papers, there are statistical estimates and analyses of the temporal variability of integral domains with certain properties. See please references 37-39:
37) (Zhmur V. V. et al. 2023) Zhmur V. V., Belonenko T. V., Novoselova E. V., Suetin B. P. Application to the World Ocean of the The-ory of Transformation of a Mesoscale Vortex into a Submesoscale Vortex Filament When the Vortex Is Elongated by an Inhomoge-neous Barotropic Flow // Oceanology. 2023c. Vol. 63, N 2, P. 184–194. https://doi.org/10.1134/S0001437023020157.
38) (Zhmur V. V. et al. 2023) Zhmur V. V., Belonenko T. V., Novoselova E. V., Suetin B.P. Conditions for Transformation of a Mesoscale Vortex into a Submesoscale Vortex Filament When the Vortex Is Stretched by an Inhomogeneous Barotropic Flow // Oceanology. 2023b. Vol. 63, N 2, P. 174–183. https://doi.org/10.1134/S0001437023020145.
39) (Zhmur V. V. et al. 2023) Zhmur V. V., Belonenko T. V., Novoselova E. V., Suetin B.P. Direct and Inverse Energy Cascades in the Ocean during Vortex Elongation // Doklady Earth Sciences. 2023a. Vol. 508, Part 2. P. 233–236. https://doi.org/10.1134/S1028334X22601675.
(4) The theory is tailored to barotropic flow. However, in the realistic scenario, both barotropic and baroclinic flows exist. Please explain the applicability of this theory to realistic cases.
The vortex is "sensitive" to the background flow field on scales of the order of the size of the vortex itself (by the thickness of the vortex above and below the core in the vertical direction and by the diameter in the horizontal direction). Outside of this volume, the background flow field can be any (Zhmur, 2011). Therefore, the concept of ‘barotropic flow’ in this case is very conditional. As for baroclinic currents, no research has been conducted in relation to our task.
Comments on the Quality of English Language
I strongly recommend the authors edit their manuscript. Right now, the English writing is of relatively poor quality, which makes it a bit challenging to read through.
Thank you for your recognition. We checked the text carefully using the available services and tried to do the best to improve language in the revised manuscript. We also revised the manuscript according to the suggestions of the professional editor.

Round 2
Reviewer 2 Report
Comments and Suggestions for Authors
Now I recommend publication.